# High SARS-CoV-2 load in the nasopharynx of patients with a mild form of COVID-19 is associated with clinical deterioration regardless of the hydroxychloroquine administration

**Alexey Komissarov**[1,2], **Ivan Molodtsov**[1,3], **Oxana Ivanova**[1,2], **Elena Maryukhnich**[1,2], **Svetlana Kudryavtseva**[1], **Alexey Mazus**[4], **Evgeniy Nikonov**[5], **Elena Vasilieva**[1,2]*

**1** Clinical City Hospital named after I.V. Davydovsky, Moscow Department of Healthcare, Moscow, Russia, **2** Laboratory of Atherothrombosis, Cardiology Department, Moscow State University of Medicine and Dentistry, Moscow, Russia, **3** N.F. Gamaleya Federal National Research Centre for Epidemiology and Microbiology, Moscow, Russia, **4** Moscow City Center for AIDS Prevention and Control, Moscow Department of Healthcare, Moscow, Russia, **5** Moscow Department of Healthcare, Moscow, Russia

* vasilievahelena@gmail.com

## Abstract

Because of the constantly growing numbers of COVID-19 infections and deaths, attempts were undertaken to find drugs with anti-SARS-CoV-2 activity among ones already approved for other pathologies. In the framework of such attempts, in a number of *in vitro*, as well as *in vivo*, models it was shown that hydroxychloroquine (HCQ) has an effect against SARS-CoV-2. While there were not enough clinical data to support the use of HCQ, several countries including Russia have included HCQ in treatment protocols for infected patients and for prophylaxis. In the current non-randomized, observational study we evaluated the SARS-CoV-2 RNA in nasopharynx swabs from infected patients 7–10 days post symptoms with clinically mild disease and compared the viral RNA load dynamics between patients receiving HCQ (200 mg twice per day according to the Ministry of Health of Russian Federation treatment instructions, $n = 33$) and a control group without antiviral pharmacological therapy ($n = 12$). We found a statistically significant relationship between maximal RNA quantity and deterioration of patients' medical conditions, and as well we confirmed arterial hypertension to be a risk factor for people with COVID-19. However, we showed that at the dose used in the study HCQ therapy neither shortened the viral shedding period nor reduced the virus RNA load.

## Introduction

Currently, the number of SARS-CoV-2 infection cases worldwide has exceeded 29 million, including more than 930 thousand registered deaths. Because of the constantly growing numbers of infections and deaths, it is paramount to find efficient antivirals that block SARS CoV-

**Data Availability Statement:** All relevant data are within the manuscript and its Supporting information files.

**Funding:** This study was funded by the Russian Science Foundation (https://www.rscf.ru/en/) grant, agreement #18-15-00420 (EV). The funder had no role in study design, data collection and analysis, decision to publish, or preparation of the manuscript.

**Competing interests:** The authors have declared that no competing interests exist.

2 infection. Since *de novo* drug development is a very long and complex process, attempts were undertaken to find drugs with anti-SARS-CoV-2 activity among ones already approved for other pathologies. Within the framework of such attempts, in a number of *in vitro* as well as *in vivo* models it has been shown that a well-known and widely available compound, hydroxychloroquine (HCQ), has an effect against SARS-CoV-2 [1–3]. Early in the endemic, there were insufficient clinical data to support a particular dose of HCQ so several countries including Russia recommended different HCQ doses as treatment for infected patients and as prophylaxis. Initial results suggesting that HCQ and chloroquine had antiviral activity and were beneficial [4–6] were not confirmed in the large Solidarity and Recovery randomized trials of severe COVID-19 [7, 8] and early treatment of mild COVID-19 [9, 10]; these trials used higher HCQ doses than the one recommended in Russia. Here, we evaluated SARS-CoV-2 RNA in nasopharynx swabs from infected patients in mild conditions and compared the viral RNA load dynamics between patients receiving HCQ ($n = 33$) and a control group without antiviral pharmacological therapy ($n = 12$). In accordance with the instructions of the Ministry of Health of the Russian Federation, patients were receiving HCQ at a dose of 200 mg twice per day. We found viral RNA load change dynamics similar to that reported in other studies [11–14], but there was no statistically significant difference between the groups.

## Materials and methods

A total of 45 patients with COVID-19 in mild condition (according to the WHO «Clinical management of severe acute respiratory infection (SARI) when COVID-19 disease is suspected: Interim guidance», 13 March 2020) were enrolled in the study during the epidemic in March–May 2020, Moscow, Russia. All the patients were examined by the medical stuff of City Polyclinic № 46 of the Moscow Department of Healthcare and were recruited 7–10 days after the onset of COVID-19 symptoms when the diagnosis was clinically confirmed from PCR analysis. Patients were not included in the study if they demonstrated evidence of severe pneumonia (respiratory rate > 30 breaths/min, severe respiratory distress, or blood oxygen saturation level ≤ 95%) on the first day of administration of the drug, and/or if they had any oncological disease. Among participants included in the study, 33 were receiving hydroxychloroquine (200 mg twice a day according to the treatment-methods instructions of the Ministry of Health of Russian Federation), while 12, who declined to take HCQ, represented a control group. The study protocol was approved by the Interuniversity Committee of Ethics, and all participants provided their written informed consent.

All the patients were regularly examined by a doctor, and the severity of symptoms was registered. Patients with body temperature holding higher than 38˚C for 4 days or more and/or with blood oxygen saturation level dropping lower than 95% were considered to be in deteriorating condition; these conditions were considered to be indications for hospitalization. However, some patients with deteriorating condition refused to be hospitalized because their subjective condition was estimated as mild. In this case the patient underwent intense home observation. Thus, two types of bad outcomes were registered and included in the analysis: condition deteriorating with and without hospitalization, referred to as «hospitalization» and «condition deteriorating» cases, respectively. At first visit, a nasopharynx swab and peripheral blood were collected from each patient; then nasopharynx swabs were collected at days 3 and 8.

Peripheral blood collected from the forearm vein into S-Monovette 2.7-mL K3E and 4.9-mL Z tubes (Sarstedt, Germany) was analyzed for complete blood count and biochemical panel, respectively, using automated procedures.

**Table 1. Oligonucleotides used for quantitative real-time PCR analysis.**

| N gene region* | Forward primer | Reverse primer | Probe |
|---|---|---|---|
| N2 | TTACAAACATTGGCCGCAAA | GCGCGACATTCCGAAGAA | FAM-ACAATTTGCCCCCAGCGCTTCAG-BHQ1 |
| N3 | GGGAGCCTTGAATACACCAAAA | TGTAGCACGATTGCAGCATTG | VIC-ATACACATTGGCACCCGCAATCCTG-BHQ2 |

*Sequences used are recommended for SARS-CoV-2 detection by Division of Viral Diseases, National Center for Immunization and Respiratory Diseases, Centers for Disease Control and Prevention, Atlanta, GA, USA. All oligonucleotides were synthesized with DNK-sintez (Russia). Sequences are presented in 5′-3′ direction. FAM, fluorescein; VIC, 2′-chloro-7′phenyl-1,4-dichloro-6-carboxy-fluorescein; BHQ1/2, Black Hole Quencher 1/2. The working concentration of each oligonucleotide is 500 nM.

The swabbing technique used was as recommended in «Interim Guidelines for Collecting, Handling, and Testing Clinical Specimens for COVID-19» by the Centers for Disease Control and Prevention, USA. Briefly, a swab with a flexible shaft was inserted through the nostril parallel to the palate until the contact with the nasopharynx and gently rolled 3 times clockwise. Specimens were collected from both nostrils using the same swab. Specimens from all patients included in the study were collected by the single medical specialist. After probing, the swabs were placed into viral transport media (COPAN Diagnostics, USA), transported at 4˚C, then stored at -20˚C. Swabs from the same patient were taken into the same volume of media, although between the patients the volume varied from 1 to 3 mL. Viral RNA was isolated using the RIBO-prep kit (AmpliSens, Russia) according to the manufacturer's protocol. Briefly, we lysed thawed transport media (100 μl), then precipitated nucleic acids using centrifugation, washed the pellet, and finally dissolved it in 50 μl of nuclease-free water. Next, 5 μl of the resulting solution was mixed with 5 μl of primer/probe mix (Table 1) and 10 μl of qScript XLT One-Step RT-qPCR ToughMix (Quantabio, USA) and was analyzed using the CFX96 Touch real-time PCR detection system (Bio-Rad, USA). The PCR program was performed as follows: 15 min at 50˚C for reverse transcription reaction followed by 5 min at 95˚C, then 50 cycles, each comprising 20 s at 95˚C, 20 s at 58˚C, and 30 s at 72˚C. Amplification of two different regions (N2 and N3) of the SARS-CoV-2 nucleocapsid (N) gene was analyzed in duplicates for each sample. For SARS-CoV-2 RNA copy number estimation, serial 10-fold dilutions of standard samples with known concentrations were used for standard curve generation, and the linear relationship between Ct values and amplicon copy number was observed for both PCR systems with the lower limit of detection being 1000 copies in PCR reaction (Fig 1). Synthetic DNA fragments containing N2 and N3 regions and viral genomic RNA (kindly provided by Inna Dolzhikova, N.F. Gamaleya Research Center for Epidemiology and Microbiology, Moscow, Russia) were used for generation of standard samples. SARS-CoV-2 RNA copy number in samples was calculated as the mean of four measurements (two values for N2 and two for N3) and extrapolated to the total volume of the transport media, thus reflecting the total viral RNA quantity in the swab. Each PCR plate contained negative control samples without matrix RNA/DNA that served as indicators of the absence of contamination.

Statistical analysis was performed with the Python3 programming language with *numpy*, *scipy*, and *pandas* packages. The Fisher exact test (two-tailed) was used for comparing qualitative parameters between independent groups of patients; the significance level α for *p*-values was set to 0.05. The Mann–Whitney U test (two-sided) was used for comparing distributions of quantitative parameters between independent groups of patients. To control for type I error, we calculated false discovery rate *q*-values using the Benjamin–Hochberg (BH) procedure, and we set a threshold of 0.05 to keep the positive false discovery rate below 5%. The Python3 package *scikit-learn* was used for plotting ROC-curves and AUC estimation.

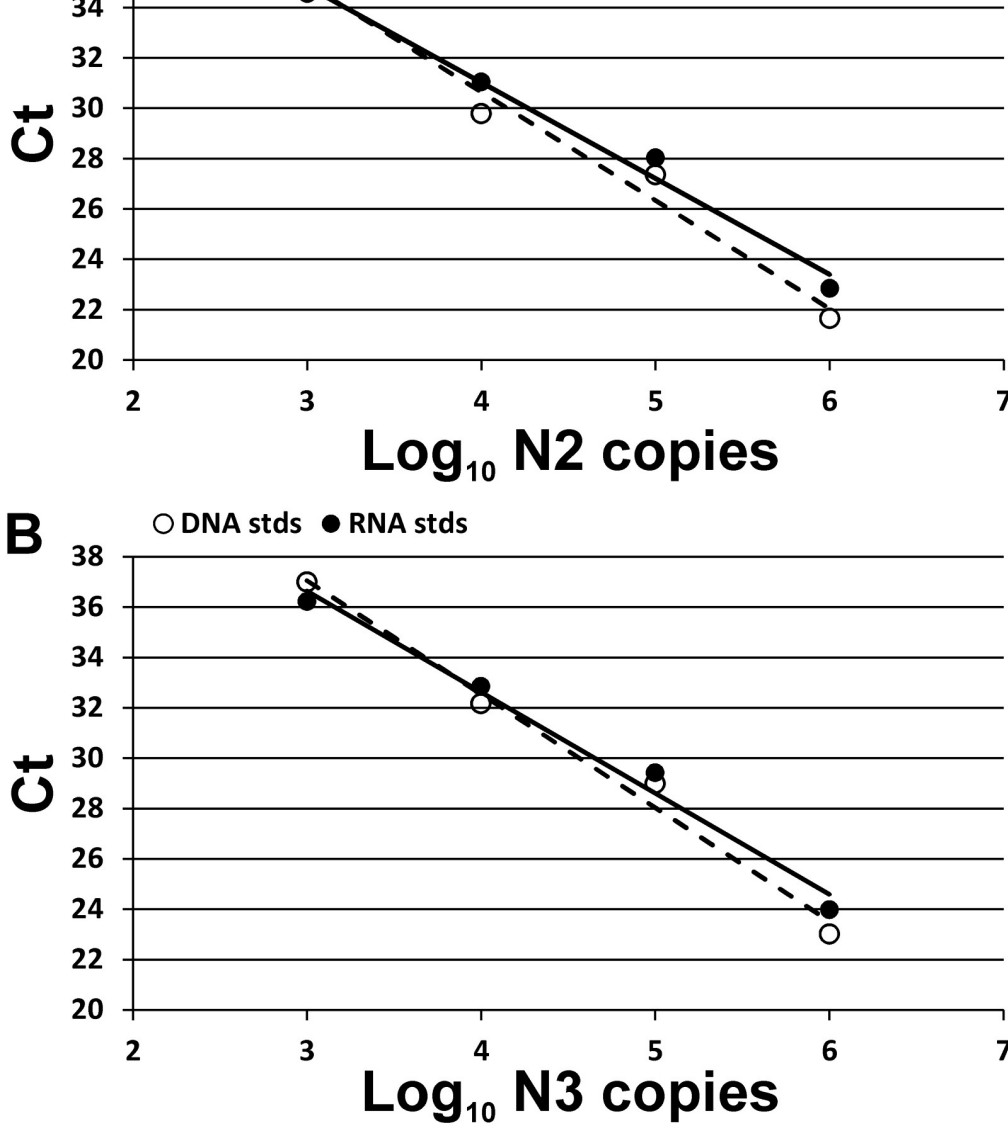

**Fig 1. Standard curves used for calculating the SARS-CoV-2 RNA absolute number in swabs.** Two PCR systems for detection of the N2 (A) or N3 (B) regions of SARS-CoV-2 nucleocapsid (N) gene were used. The curves were generated using standard samples with either synthetic DNA fragments (white dots, dashed lines) or viral genomic RNA (black dots, solid lines). Both standards demonstrate similar results.

## Results and discussion

In the current study, 45 patients with COVID-19 clinically confirmed by PCR, in mild condition (according to the WHO COVID-19 disease severity classification, «Clinical management of severe acute respiratory infection (SARI) when COVID-19 disease is suspected: Interim guidance», 13 March 2020), were analyzed for SARS-CoV-2 RNA in the nasopharynx. On the initial day of the study ("day 0"), peripheral blood was collected and the complete blood count

**Table 2. General characteristics of experimental groups.**

| Parameter | Control (*n* = 12) | | HCQ (*n* = 33) | | *p*-value |
|---|---|---|---|---|---|
| Males/Females, % | | 17/83 | | 60/40 | NA |
| Age, years | 53.0 | [44.8;60.8] | 49.0 | [38.0;57.0] | 0.23 |
| Comorbidities: | | | | | |
| Diabetes, n (%) | | 0 (0) | | 3 (9) | NA |
| Hypothyrosis, n (%) | | 1 (8) | | 1 (3) | NA |
| Arterial hypertension, n (%) | | 3 (25) | | 13 (39) | NA |
| Time between the onset of symptoms and inclusion, days | | 9 [8.0;9.0] | | 8.5 [8.0;9.0] | 0.91 |
| Hemoglobin, g/L | 124.0 | [123.0;132.0] | 141.0 | [131.0;151.0] | 0.079 |
| Erythrocytes, $10^{12}$/L | 4.4 | [4.3;4.6] | 4.8 | [4.4;5.0] | 0.16 |
| Mean corpuscular volume, fL | 86.0 | [84.0;88.0] | 89.0 | [87.0;91.0] | 0.086 |
| Mean corpuscular hemoglobin, pg | 29.6 | [27.8;29.8] | 30.0 | [29.6;30.9] | 0.087 |
| Red blood cell distribution width, % | 14.1 | [13.3;14.1] | 12.6 | [12.4;13.1] | **0.032** |
| Mean corpuscular hemoglobin concentration, g/dL | 33.8 | [33.2;34.0] | 34 | [33.6;34.3] | 0.23 |
| Platelets, $10^9$/L | 239.0 | [223.0;309.0] | 231 | [173.0;268.0] | 0.15 |
| Leukocytes, $10^9$/L | 6.9 | [5.1;8.5] | 5.1 | [4.4;6.3] | 0.079 |
| Neutrophils, % | 61.4 | [59.0;65.7] | 52.3 | [46.5;59.9] | 0.087 |
| Lymphocytes, % | 27.7 | [26.0;32.0] | 39.9 | [30.9;44.3] | 0.087 |
| Monocytes, % | 7.0 | [3.5;8.6] | 6 | [4.9;6.6] | 0.31 |
| Eosinophils, % | 2.0 | [1.5;2.3] | 1.4 | [1.1;1.8] | 0.15 |
| Basophils, % | 0.5 | [0.5;0.6] | 0.5 | [0.4;0.6] | 0.34 |
| Creatinine, µM | 91.0 | [80.0;100.0] | 92 | [84.0;111.0] | 0.27 |
| Total cholesterol, mM | 4.6 | [4.2;5.7] | 3.5 | [3.1;4.0] | **0.048** |
| Triglycerides, mM | 1.5 | [1.0;1.8] | 1.3 | [0.9;1.8] | 0.45 |
| HDLP, mM | 1.1 | [0.9;1.2] | 0.8 | [0.7;1.1] | 0.079 |
| LDLP, mM | 3.2 | [2.4;3.8] | 1.9 | [1.4;2.2] | 0.079 |
| Total bilirubin, µM | 7.5 | [5.4;9.7] | 10.3 | [8.4;12.3] | 0.11 |
| Direct bilirubin, µM | 1.7 | [1.2;1.8] | 1.8 | [1.4;2.2] | 0.26 |
| Indirect bilirubin, µM | 5.8 | [4.2;7.4] | 9 | [6.7;10.6] | 0.087 |
| Alanine aminotransferase, IU/L | 21.0 | [17.0;24.0] | 27 | [21.0;38.0] | 0.15 |
| Aspartate aminotransferase, IU/L | 17.0 | [15.0;19.0] | 16 | [11.0;22.0] | 0.45 |
| Glucose, mM | 5.6 | [5.0;7.5] | 5.8 | [5.3;6.3] | 0.38 |
| C-reactive protein, mg/L | 0.1 | [0.1;5.6] | 3.3 | [0.1;17.4] | 0.25 |

HCQ, hydroxychloroquine. Values presented as median [quartile 2; quartile 3]. Differences between groups were quantified using the Mann–Whitney U test with the Benjamini-Hochberg correction. Significant differences ($p < 0.05$) are marked in bold. NA, *p*-level cannot be calculated.

as well as the biochemical panel were analyzed. Nasopharynx swabs were taken on days 0, 3, and 8 after inclusion in the study, corresponding respectively to 7–10, 10–13, and 15–18 days after the onset of COVID-19 symptoms. Among patients included in the study, 33 were receiving HCQ while 12 were receiving no antiviral pharmacological therapy. The age ranges of patients in the control and experimental groups were comparable, but the groups differed in sex ratio (Table 2). Nevertheless, the key complete blood parameters and biochemical characteristics were indistinguishable between the groups except for red blood cell distribution width (RDW, $p < 0.05$) and total cholesterol (TH, $p < 0.05$); however, all the parameters including RDW and TH were in the normal range (Table 2).

We found that the total viral RNA quantity in nasopharynx swabs of all patients included in the study on day 0 was ranging from $10^4$ to $10^9$ copies, with median and interquartile range

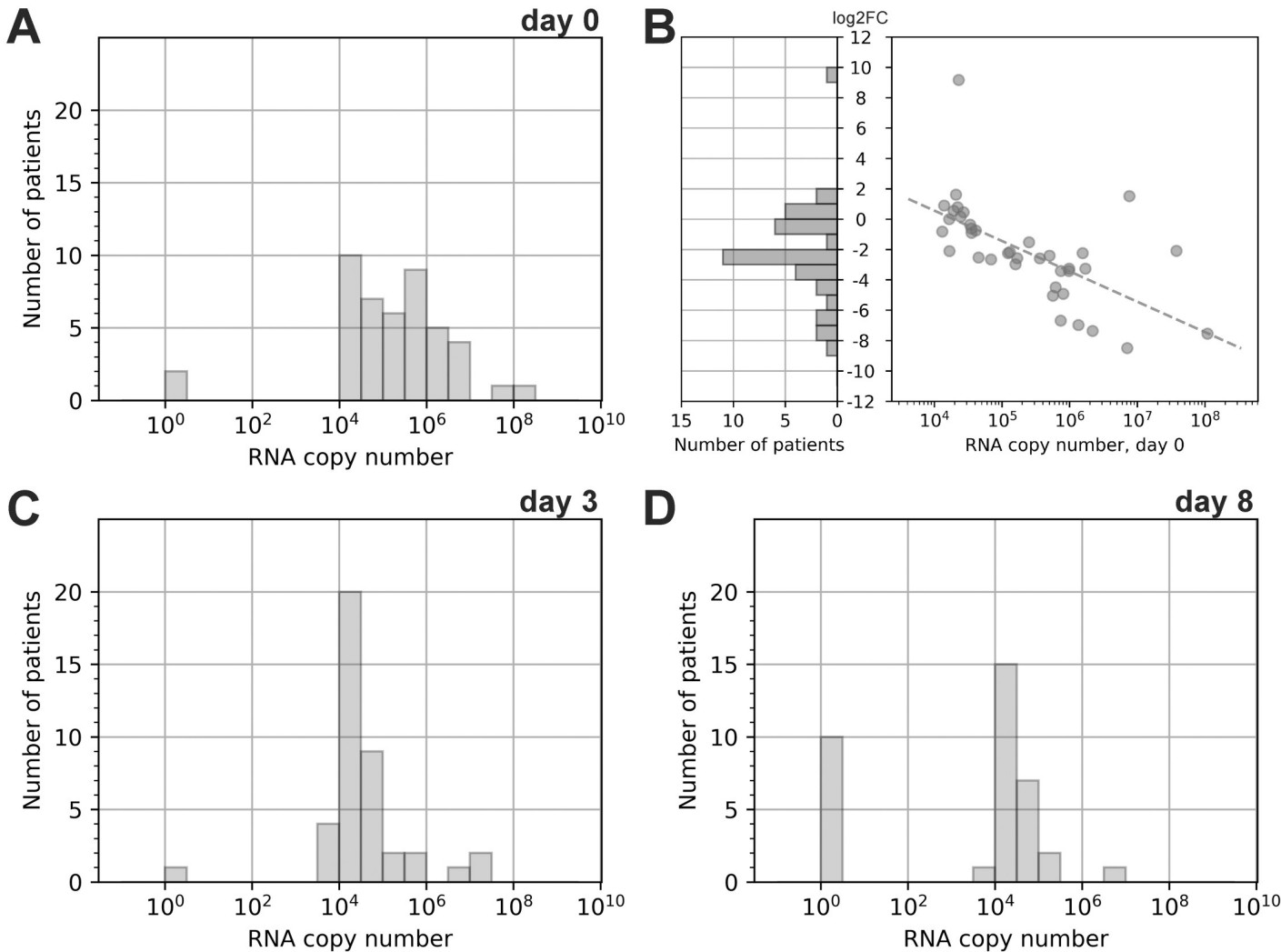

**Fig 2. SARS-CoV-2 RNA quantity change dynamics.** The distribution of viral RNA load in the nasopharynx of patients at day 0 (A), day 3 (C), and day 8 (D). To use the logarithmic scale, the exact 0 was replaced with 1 ($10^0$). Distribution of log2 transformed fold change of RNA copy number between swabs on days 0 and 3 (B Left). Dependence between log2 transformed fold change of RNA copy number and RNA copy number on day 0; dashed line shows linear approximation of the trend (Spearman correlation, $r$ = -0.7, $p$-value < 0.00001) (B Right). Only patients having non-zero values of RNA copy number on both day 0 and day 3 and were not hospitalized between days 0 and 3 are shown.

150 000 [25 000 – 1 000 000]. It is noteworthy that for two patients we obtained negative results. The distribution of RNA copies had a maximum shifted towards 10 000–20 000 copies and a prolonged right shoulder (Fig 2A). Measurement of viral RNA in the course of the disease revealed that on day 3 for most of the patients (68.9%, 31 patients out of 45) RNA copies decreased, mostly more than 4-fold (53.3%, 24 patients out of 45). Meanwhile, the increase in RNA load was observed in 22.2% (10 of 45) patients, in some of them (6.7%, 3 of 45) more than 4-fold. Furthermore, we found that higher RNA quantity on day 0 was correlated with greater fold change between days 0 and 3 (Fig 2B). This, together with the elimination of a significant proportion of patients with high viral load due to hospitalization, resulted in a significant narrowing of the RNA copies range at day 3 and its "compression" around the value of ~10 000. We found an even narrower RNA load distribution on day 8 of the study with 22.2% (10 of 45) of patients having negative swabs (Fig 2D). We believe that this distribution resulted

from a delay between the onset of the symptoms and the patient's inclusion in the study. Because of the acute shortage of PCR tests during the epidemic, nasopharynx swabs for clinical COVID-19 confirmation were taken at the day of symptom onset, but the results of the PCR-analysis were available only 7–10 days later. Thus, several patients may have already recovered from the infection during these days, resulting in low RNA copy numbers or negative results even on the first day of the study.

Analyzing individual RNA load dynamics among patients who neither demonstrated severe symptoms during the disease nor were hospitalized, we found that these patients were characterized either by RNA loads of $10^4$ at the start of the study or by RNA load parameters that had dropped to this value by day 3 or 8. This observation may indicate that in our experimental system RNA quantity in the order of $10^4$ is associated with the recovery stage of infection. This assumption is supported by the fact that at the time when the RNA load dropped to the $10^4$ patients were characterized by normal body temperature and demonstrated no symptoms of the disease; however, additional studies are needed to prove this assumption.

The results of our study are in agreement with previously published data, from which it was found that throat and nasal swabs of infected individuals are characterized by RNA copy numbers varying from $10^2$ to $10^{11}$ [11–14]. Although direct comparison between the results of these studies and our work is complicated because of differences in experimental procedures, similar trends were observed, e.g., recovering patients demonstrated considerably lower RNA copy numbers compared with the productive stage of infection, and higher viral load is associated with more adverse symptoms. Additionally, the results of the current study indicate that recovering patients may produce viral RNA even 18 days after the onset of symptoms. Similar results were demonstrated by Zhou F et al. [15] for patients with COVID-19 in Wuhan, China–in this study the median duration of viral shedding was found to be 20 days (IQR 17–24), with the longest observed duration of viral shedding being 37 days.

Comparison of viral RNA dynamics in individual groups (S1A Fig) showed that there were no statistically significant correlations between decrease/increase in RNA copy number and the administration of HCQ–at each time point the RNA load in swabs was comparable between the groups (Table 3 and S1B Fig). Moreover, on day 8 (end of the observation period in the current study) 33.3% (4 of 12) and 18% (6 of 33) of patients were characterized by negative swabs in the control and the HCQ-receiving groups, respectively. Taken together, these results demonstrate that HCQ therapy neither shortened the viral shedding period nor reduced the virus RNA load.

Further analysis has shown no statistically significant correlations between maximal RNA quantity in swabs and patients' age, sex, or blood parameters. However, we found a significant relationship between patients' RNA quantity and both deteriorating medical conditions and hospitalization. To estimate the ability of viral RNA load to predict bad outcomes, corresponding ROC-curves were analyzed (S1C Fig). Accordingly, any threshold in the range from

**Table 3. Comparison of viral RNA load at different time points between groups.**

|  | Control group | | HCQ group | | Mann-Whitney U Test | |
|---|---|---|---|---|---|---|
|  | n | median IQR[Q2;Q3] | n | median IQR[Q2;Q3] | p-value unadj | q-value |
| Day 0, total RNA copies in swab | 12 | 447 847 [23 821;1 030 282] | 33 | 130 183 [33 812;982 481] | 0.38 | 0.45 |
| Day 3, total RNA copies in swab | 12 | 27 638.5 [20 138;45 794] | 29 | 27 715.0 [17 204;87 433] | 0.45 | 0.45 |
| Day 8, total RNA copies in swab | 10 | 19 001.5 [0;28 313] | 26 | 23 880.5 [8 094;36 859] | 0.18 | 0.45 |

HCQ, hydroxychloroquine; IQR, interquartile range; Q2, quartile 2; Q3, quartile 3. Differences between groups were quantified using the Mann–Whitney U test without (p-value unadj) and with the Benjamini-Hochberg correction (q-value).

**Table 4. Significant correlations between RNA quantity and patients' clinical data.**

|  | no deterioration | deterioration | *p*-value |
|---|---|---|---|
| RNA copy number $\leq 10^6$ | 26 | 6 | 0.000066 |
| RNA copy number $> 10^6$ | 2 | 11 |  |
|  | **non-hospitalized** | **hospitalized** | ***p*-value** |
| RNA copy number $\leq 10^6$ | 29 | 3 | 0.0029 |
| RNA copy number $> 10^6$ | 6 | 7 |  |
|  | **non-hospitalized** | **hospitalized** | ***p*-value** |
| no arterial hypertension | 27 | 8 | 0.0091 |
| arterial hypertension | 3 | 7 |  |

*p*-values were determined using Fisher's exact test.

$1.2 \times 10^5$ to $1.2 \times 10^6$ is characterized by the same FPR/TPR values. Thus, RNA load equal to $10^6$ copies has been chosen arbitrary for demonstration purposes; however, further studies with more patients will be required to specify the precise threshold value. We found that patients with RNA loads higher than $10^6$ copies showed deterioration in their condition significantly more frequently than those with RNA loads below $10^6$ copies ($p = 0.000066$, Fisher exact test) (Table 4). Among hospitalized patients, only one belonged to the control group while the rest (9 of 10) were receiving HCQ; however, this difference may be the result of the smaller size of the control group. Hospitalization cases also showed a strong positive correlation with viral RNA load: patients with higher than $10^6$ viral RNA copies were hospitalized significantly more frequently than those with RNA loads below $10^6$ copies ($p = 0.0029$, Fisher exact test) (Table 4). The ages and blood parameters of patients were comparable between the hospitalized and non-hospitalized groups. However, in agreement with recently published data [16], we found that patients with arterial hypertension were hospitalized significantly more frequently than patients without hypertension ($p = 0.009$, Fisher exact test) (Table 4), though there was no correlation between this parameter and viral RNA load.

## Conclusions

We found that there is a statistically significant positive correlation between SARS-CoV-2 RNA quantity in the nasopharynx and deterioration of patients' medical conditions leading to hospitalization. At the dose used in this study in mildly ill patients who were symptomatic for at least one week, HCQ did not accelerate viral clearance compared to no HCQ administration. Although our study has a significant limitation due the relatively small number of patients, our findings, together with the results of recently published works, indicate that quantitative PCR can be a prospective approach for monitoring of COVID-19 course and prediction of deterioration in patient condition.

## Supporting information

**S1 Fig. SARS-CoV-2 RNA quantity comparison.** Individual dynamics (A) and intergroup comparison (B) of measured RNA copy number. Each curve represents a patient. To use the logarithmic scale, the exact 0 was replaced with 1 ($10^0$). Shown are ROC-curves and corresponding area under the curve (AUC) values reflecting the sensitivity and specificity of different viral RNA load values used to predict outcomes with clinical deterioration or hospitalization (C).
(TIF)

## Acknowledgments

We thank Dr. Inna Dolzhikova (N.F. Gamaleya Research Center for Epidemiology and Micro-biology, Moscow, Russia) for providing with SARS-CoV-2 genomic RNA used for the calcula-tion of absolute number of viral RNA in samples.

## Author Contributions

**Conceptualization:** Alexey Mazus, Evgeniy Nikonov, Elena Vasilieva.

**Formal analysis:** Ivan Molodtsov.

**Funding acquisition:** Elena Vasilieva.

**Investigation:** Alexey Komissarov, Ivan Molodtsov, Oxana Ivanova, Elena Maryukhnich, Sve-tlana Kudryavtseva.

**Methodology:** Alexey Komissarov, Svetlana Kudryavtseva.

**Project administration:** Alexey Komissarov.

**Resources:** Evgeniy Nikonov.

**Supervision:** Alexey Mazus, Evgeniy Nikonov, Elena Vasilieva.

**Writing – original draft:** Alexey Komissarov.

**Writing – review & editing:** Ivan Molodtsov, Elena Vasilieva.

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
