## [Decision Letter · Decision Letter 0]

13 Nov 2020

PONE-D-20-29297

Hydroxychloroquine has no effect on SARS-CoV-2 load in nasopharynx of patients with mild form of COVID-19

PLOS ONE

Dear Dr. Vasilieva, 

Thank you for submitting your manuscript to PLOS ONE. After careful consideration, we feel that it has merit but does not fully meet PLOS ONE’s publication criteria as it currently stands. Therefore, we invite you to submit a revised version of the manuscript that addresses the points raised during the review process.

Please review the comments and respond accordingly.

We look forward to receiving your revised manuscript.

Kind regards,

Walter R. Taylor

Academic Editor

PLOS ONE

Additional Editor Comments (if provided):

Dear Dr. Vasilieva,

we have received the comments from one external reviewer and I too have reviewed it in detail.

Overall, your paper is interesting but needs to undergo a lot of revision.

yours sincerely,

Water Taylor

Journal Requirements:

2. Please provide a sample size and power calculation in the Methods, or discuss the reasons for not performing one before study initiation.

3. In your Methods section, please provide additional information about the participant recruitment method and the demographic details of your participants. Please ensure you have provided sufficient details to replicate the analyses such as: a) the recruitment date range (month and year), b) a description of any inclusion/exclusion criteria that were applied to participant recruitment, c) a table of relevant demographic details, d) a description of how participants were recruited, and e) descriptions of where participants were recruited and where the research took place.

4. Please provide additional details regarding participant consent. In the ethics statement in the Methods and online submission information, please ensure that you have specified whether consent was informed. If your study included minors, state whether you obtained consent from parents or guardians. If the need for consent was waived by the ethics committee, please include this information.

5.Thank you for including your ethics statement:  "Local ethics committees approved the study protocol and all participants provided their written consent.".   

**Comments to the Author**

1. Is the manuscript technically sound, and do the data support the conclusions?

Reviewer #1: Partly

2. Has the statistical analysis been performed appropriately and rigorously? 

Reviewer #1: No

3. Have the authors made all data underlying the findings in their manuscript fully available?

Reviewer #1: Yes

4. Is the manuscript presented in an intelligible fashion and written in standard English?

Reviewer #1: Yes

5. Review Comments to the Author

Reviewer #1: Vasileva et al. paper.

Hydroxychloroquine has no effect on SARS-CoV-2 load in nasopharynx of patients with mild form of COVID-19

Thank you for submitting the manuscript and congratulations in being able to complete research during the COVID-19 pandemic, with all the inherent difficulties of this.

The main concern with this paper as it does not robustly show what it reports to in the title i.e. the authors have not demonstrated that Hydroxychloroquine has no effect on SARS-CoV-2 in mild COVID-19, but instead that in this study no effect was seen, and as there are methodological shortcomings identified below, I do not feel the strong conclusions are supported by the supplied data. A far more interesting observation, which perhaps should be the topic of the paper was that maximal RNA quantity predicted clinical outcome.

Issues with study design are (Major):

• Dose- this is based on modelling much too low, being well below modelled parameters which are safe and given the mild antiviral effect of this drug, likely to be effective. How was the dose determined?

• Timing of intervention- enrolling people at 7-10 days is likely too late for an effect and is in the time range of the RECOVERY and SOLIDARITY trials where HCQ was shown to definitively be clinically ineffectual. Studies have shown that viral loads peak at around symptom onset, and decrease monotonically thereafter. In contrast, clinical deterioration occurs later, and based on the beneficial effects of steroids in late-stage illness, is likely not directly related to virus but more likely inflammation (as steroids worsen viral replication). RCTs conducted in earlier treatment have shown some clinical benefit and this would be a more interesting group in which to conduct the study. The delayed initiation of antiviral therapy significantly decreases the a priori likelihood of a beneficial effect, and means that maximal RNA may have not been captured, but a delayed clearance of virus instead.

• Number and choice of participants. Numbers were low and it is not clear how participants were selected to receive HCQ or control. If these patients had been selected to receive different treatments by a treating physician, as opposed to a randomised manner, this allows bias to enter. No mention of baseline characteristics of participants was mentioned other than sex, age and blood parameters and other clinical factors, as well as duration from symptom onset, should be addressed.

• Swabbing frequency and technique: although there is good discussion of the PCR techniques used, it is not clear how the swabbing frequency of D0, 3 and 8 were chosen and the technique used. Additionally, as the inter-swab variation clearly creates noise for the result, it would be good to see how the swabbing frequency and the sample size was initially determined, as well as the null hypothesis which would could be rejected. Without these considerations, it seems that the statistical analysis is post hoc.

• ‘Mild’ is not defined. Not clear what definition was used here and this would be significant, in particular if HCQ/ placebo were initiated based on Physician judgement.

Minor issues:

• Grammar. Would benefit from a review by a native English-speaker. E.g. “Hydroxychloroquine has no effect on SARS-CoV-2 load in nasopharynx of patients with a mild form of COVID-19.”

In summary, the more interesting result of the paper is that maximal RNA quantity predicted clinical outcome, although some of the issues relating to study design also temper the strength of this conclusion. I do not feel that the study, as was conducted, was able to support the strong conclusion that “Hydroxychloroquine has no effect on SARS-CoV-2 load in nasopharynx of patients with a mild form of COVID-19”, for the reasons of dose, timing, patient selection (prone to bias) and statistical considerations around study power.

6. PLOS authors have the option to publish the peer review history of their article (what does this mean?). If published, this will include your full peer review and any attached files.

Reviewer #1: No

---

## [Author Response · Author response to Decision Letter 0]

2 Dec 2020

First, together with co-authors we would like to thank the reviewers for thorough examination of our manuscript and helpful notes and suggestions. The amendments introduced are listed below:

Response to the reviewer’s comments:

Issues with study design are (Major):

“Dose- this is based on modelling much too low, being well below modelled parameters which are safe and given the mild antiviral effect of this drug, likely to be effective. How was the dose determined?”

The hydroxychloroquine dose used in the study (200 mg twice per day) was the one recommended by the Ministry of Health of Russian Federation as prophylaxis and treatment option for COVID-19. This information has been added to the text (Page 2, Lines 28-30; Page 3, Lines 50-53, 65-66).

“Timing of intervention- enrolling people at 7-10 days is likely too late for an effect and is in the time range of the RECOVERY and SOLIDARITY trials where HCQ was shown to definitively be clinically ineffectual. Studies have shown that viral loads peak at around symptom onset, and decrease monotonically thereafter. In contrast, clinical deterioration occurs later, and based on the beneficial effects of steroids in late-stage illness, is likely not directly related to virus but more likely inflammation (as steroids worsen viral replication). RCTs conducted in earlier treatment have shown some clinical benefit and this would be a more interesting group in which to conduct the study. The delayed initiation of antiviral therapy significantly decreases the a priori likelihood of a beneficial effect, and means that maximal RNA may have not been captured, but a delayed clearance of virus instead.”

During the COVID-19 epidemic in the spring of 2020 there was the acute shortage of PCR tests due to the absence of the developed diagnostic PCR-systems for SARS-CoV-2 detection. In this regard nasopharynx swabs for clinical COVID-19 confirmation were taken at the day of the symptoms onset, but the results of the PCR-analysis were available only 7-10 days after. Accordingly, hydroxychloroquine administration started only after the COVID-19 confirmation. Thus, in the current circumstances at that time patients were included into the study with the 7-10 days delay after the symptoms onset. Apparently, this situation resulted in specific shape of RNA load distribution that we observed and in negative PCR-results for two patients at first day of the study. In this respect we agree that the results of our study are applicable to a later stage of infection rather than its onset. Still, we believe that the data obtained may provide the valuable information for the more detailed understanding the course of SARS-CoV-2 infection. We expanded the discussion of this issue in the text and emphasized that the results of the study were obtained for the 7-18 days after the onset of the COVID-19 symptoms (Page 3, Lines 59-61; Page 7, Lines 142-143; Page 9, Lines 169-174).

“Number and choice of participants. Numbers were low and it is not clear how participants were selected to receive HCQ or control. If these patients had been selected to receive different treatments by a treating physician, as opposed to a randomised manner, this allows bias to enter. No mention of baseline characteristics of participants was mentioned other than sex, age and blood parameters and other clinical factors, as well as duration from symptom onset, should be addressed.”

If COVID-19 in mild form is clinically confirmed the patient is recommended to start taking the HCQ in accordance with the methodical instructions of the Ministry of Health of Russian Federation. However, some patients refused this recommendation because of subjective reasons. In this case these patients were included into the study in the control group if they provided their written consent. Thus, a possibility of existing of the bias is present since inclusion into the groups was not strictly randomized; however, we compared the baseline characteristics of participants between the groups, including blood parameters, the presence of comorbidities, duration from symptom onset, and didn’t find any statistically significant differences. These results indicate that general characteristics of participants were comparable between the groups which significantly reduces the possibility of bias existence, although it does not completely exclude it.

“Swabbing frequency and technique: although there is good discussion of the PCR techniques used, it is not clear how the swabbing frequency of D0, 3 and 8 were chosen and the technique used. Additionally, as the inter-swab variation clearly creates noise for the result, it would be good to see how the swabbing frequency and the sample size was initially determined, as well as the null hypothesis which would could be rejected. Without these considerations, it seems that the statistical analysis is post hoc.”

According to the methodical instructions of the Ministry of Health of Russian Federation if COVID-19 in mild form is clinically confirmed the patient is recommended to start taking the HCQ for 7 days. Thus, the swabbing frequency was chosen in a way to collect samples at first day (day 0), at the last day (day 8) and in the middle (day 3) of the drug administration period. Swabbing technique was used as recommended in «Interim Guidelines for Collecting, Handling, and Testing Clinical Specimens for COVID-19» by Centers for Disease Control and Prevention (https://www.cdc.gov/coronavirus/2019-ncov/lab/guidelines-clinical-specimens.html). Briefly, a swab with a flexible shaft was inserted through the nostril parallel to the palate until the contact with the nasopharynx. Next, swap was gently rolled 3 times clockwise. Specimens were collected from both nostrils using the same swab, then the swab was placed into the viral transport medium. All specimens were collected by the single medical specialist. This information has been added to the text (Page 4, Lines 85-91).

“‘Mild’ is not defined. Not clear what definition was used here and this would be significant, in particular if HCQ/ placebo were initiated based on Physician judgement.”

For the COVID-19 disease severity classification we used WHO «Clinical management of severe acute respiratory infection (SARI) when COVID-19 disease is suspected: Interim guidance», 13 March 2020. In accordance with this guidance the «mild» condition was defined as clinically confirmed COVID-19 without evidence of pneumonia or hypoxia (normal respiratory rate and blood oxygen saturation). We added this information to the text (Page 3, Lines 56-58, 61-64).

Minor issues:

“Grammar. Would benefit from a review by a native English-speaker. E.g. “Hydroxychloroquine has no effect on SARS-CoV-2 load in nasopharynx of patients with a mild form of COVID-19.””

The manuscript text was edited by a native English speaker who is a professional in the subject area of the article (Throughout the text).

“In summary, the more interesting result of the paper is that maximal RNA quantity predicted clinical outcome, although some of the issues relating to study design also temper the strength of this conclusion. I do not feel that the study, as was conducted, was able to support the strong conclusion that “Hydroxychloroquine has no effect on SARS-CoV-2 load in nasopharynx of patients with a mild form of COVID-19”, for the reasons of dose, timing, patient selection (prone to bias) and statistical considerations around study power.”

We agree that the results of the current study should not be extrapolated to all concentrations of HCQ, since we have shown that HCQ administration shows no pronounced effect on SARS-CoV-2 load only at the dose used and in patients with mild form of disease. Accordingly, we edited the text and made more accurate conclusions (Page 12, Lines 243-245). In addition, we corrected the title of the study to shift the focus to the found correlation between RNA load and clinical outcomes in accordance with the reviewer’s comments (Page 1, Lines 1-2).

Response to the Academic Editor’s comments:

Abstract:

“We need to see the dose of HCQ recommended in Russia at the time of the study and the dose given to patients in this report.”

The hydroxychloroquine dose used in the study was 200 mg twice per day and this dose is the one recommended by Ministry of Health of Russian Federation as prophylaxis and treatment option for a mild COVID-19. This information has been added to the text according to the comments (Page 2, Lines 28-30).

“Need to see the size of each group and it should be stated here that the study was a non-randomised, observational study.”

The size of each group and statements concerning the study being non-randomised and observational have been added to the text (Page 2, Lines 26, 27, 29, 30).

“The last sentence should be modified to say at the dose used, HCQ……”

The corresponding sentence was modified according to the comments (Page 2, Lines 33).

Introduction:

“The authors must mention the Recovery and Solidarity trials which showed that chloroquine and HCQ, respectively, had no effect in severe COVID-19 disease. This paper has also just been published in JAMA and the authors may wish to cite this as well: Self et al. National Heart, Lung, and Blood Institute PETAL Clinical Trials Network. Effect of hydroxychloroquine on clinical status at 14 days in hospitalized patients with COVID-19: a randomized clinical trial. JAMA. Published online November 9, 2020.

They must also mention several studies of viral kinetics on COVID-19 to give readers an idea of how viral load varies over time; the focus should be on patients with mild disease and asymptomatic individuals.”

According to the comments, corresponding papers have been cited (Page 3, Line 45).

“We must know the dose of HCQ used in Russia as treatment and prophylaxis.”

The explanation has been introduced into the text (Page 3, Lines 50-51).

Methods

“How were the patients recruited?”

All individuals included into the study were the patients of the City Polyclinic № 46 of the Moscow Department of Healthcare and were recruited if their COVID-19 was clinically confirmed by PCR analysis after their written consent had been signed. This information has been introduced into the text (Page 3, Lines 56-61).

“Were there no formal inclusion or exclusion criteria?”

During the study there was an exclusion criterion: patients were not included into the study if they demonstrated evidence of severe pneumonia and/or hypoxia at the first day of the drug administration and/or had any oncological disease. This information has been introduced into the text (Page 3, Lines 61-64).

“The authors have defined clinical deterioration but what were the clinical indications for hospitalisation? The authors may wish to refer to the WHO classification of COVID-19 severity.”

Indeed, for the COVID-19 disease severity classification we used WHO «Clinical management of severe acute respiratory infection (SARI) when COVID-19 disease is suspected: Interim guidance», 13 March 2020. In accordance with the WHO guidance and Ministry of Health of Russian Federation methodical instructions patient’s body temperature holding higher than 38 °C for 4 days or more and/or the blood oxygen saturation level dropping lower than 95% were considered as condition deteriorating and were the indications for hospitalization. However, some patients with condition deteriorating refused to be hospitalized because their subjective condition was mild. Since hospitalization without the patient’s consent is prohibited, we had two types of bad outcomes: condition deteriorating with and without hospitalization. Thus, we included both types of outcomes into the analysis. We expanded the discussion of this issue in the text (Page 3, Lines 70-77; Page 7, Lines 138-140).

“For the viral loads, the authors talk of taking into account the volume of viral transport medium so I am expecting to see viral load/unit volume but this is not reported. Please clarify.”

During the study tubes with different amount of viral transport media (1-3 mL, COPAN Diagnostics, USA) were used for nasopharynx probing. This situation was concerned with acute shortage of materials during the first wave of COVID-19 epidemic. However, swabs from the same patient were taken into the same volume of media, although between the patients volume varied from 1 to 3 mL. In this respect direct comparison of the viral loads per volume unit between samples is incorrect. So, for the comparison of viral load dynamics between patients RNA amount was calculated per total volume. This information has been introduced into the text (Page 5, Lines 91-92).

“The authors should mention here how they chose a viral load of 106 to use as a marker to differentiate those with and without clinical deterioration.”

RNA load equal to 106 copies as a classifier has been chosen arbitrary for demonstration purposes based on the analysis of the ROC-curves. We found that in our data any threshold in the range from 1.2×105 to 1.2×106 RNA copies is characterized by the same FPR/TPR values. Additional studies with more patients will be required to specify the precise threshold value. We added the discussion of this issue in the text (Figure S1,C and Page 11, Lines 218-226).

“I have never heard of a continuity correction for a Mann Whitney U test. Please remove.”

The text was edited in accordance with the comment (Page 6, Line 130).

Results & Discussion

“Clinicians will want to know that all patients had virologically-proven COVID-19. This is not clear from the results as some patients did not have a detectable RNA load. Please clarify.”

The text was edited in accordance with the comment (Page 7, Lines 137; Page 9, Lines 169-174).

“It is unclear to me regarding the patient with a zero viral load when the authors say the lower limit of detection with the qPCR is 1000 copies.”

We agree that negative PCR results does not obligatory indicate that viral RNA load is zero, since viral load may be just lower than the detection limit of our PCR systems used. We edited the text according to the comments and substitute “zero load” for “negative results” (Page 8, Lines 157-159).

“Line 147. The authors speculate that a RNA viral load of 104 may be associated with clinical recovery but no clinical data are shown.”

Analyzing individual RNA load dynamics among patients who had good outcome (i.e., neither was hospitalized nor had clinical deterioration) we found that these patients were characterized by RNA load of 104 either at the start of the study or the parameter dropped to this value by day 3 or 8. In addition, according to the medical reports of the patients observation at the time when the RNA load dropped to the 104 patients were characterized by normal body temperature and demonstrated no symptoms of the disease. Based on these data we suggested that RNA quantity around 104 is associated with the recovery stage of infection, however additional studies needed to prove this suggestion. We added the discussion of this issue to the text (Page 9, Lines 169-174; Lines 184-191).

“In describing the viral dynamics, I suggest the authors use such Figures as:

Individual viral loads over time (raw data & log transformed) with illness day on the X axis using different colours for the control and HCQ groups. This will give a much better idea of viral dynamics. If this is too busy, then show two graphs – control & HCQ.

Box plots of viral load using the raw data on D0, 3 & 8 with one plot for the control group and another for the HCQ group. This will give a visual impression to support the data in Table 3.”

According to the comments Supplementary Figure S1 has been generated.

“The authors suggest a high viral load is associated with clinical deterioration and hospital admission. This is an important finding. Have other authors found this?”

Yes, association between high viral load and more adverse symptoms has already been observed for COVID-19. We added discussion and corresponding references into the text (Page 10, Lines 193-198).

“Hypertension is mentioned as a significant factor but did the subjects have any other comorbidities?”

Patients included into the study had no other comorbidities except for arterial hypertension (16 of 45), diabetes (3 of 45) and hypothyrosis (2 of 45). This information was added to the Table 2.

“Were any patients on statins?”

Yes, two patients were taking rosuvastatin and two – atorvastatin. Three patients belonged to the HCQ group, one – to the Control group. However, because of too small number of patients on statins statistical correlations cannot be estimated.

“Although the study is small with limited statistical power, the authors may wish to use logistic regression to look at several factors for clinical deterioration and hospital admission. Similarly, for the viral load over time between the two groups, repeated measures ANCOVA (log transformed data) or mixed effects modelling could be used to compare the two groups.”

We thank the reviewer for the valuable remark. We indeed considered making more elaborate predictive models and tried performing logistic regression and other types of regressions/tests; however, small size of the cohort and low number of hospital admissions/clinical deterioration events presented a serious problem which didn't allow us to come to any conclusive results. Thus, we decided to stop on the easiest-to-observe and most reliable effects and leave other types of analysis for further studies with more subjects.

Conclusion

“All the authors can say here is that their small study of mild disease did not show an apparent effect on viral load of HCQ at the dose used. Perhaps the dose was too low? They cannot extrapolate their findings to primary prophylaxis so this should be removed.”

We absolutely agree with the reviewer comments that the results of the study are relevant only for the tested HCQ dose and mild COVID-19 conditions. Thus, the text was edited in accordance with the comment (Page 12, Lines 243-245).

---

## [Editor Report · Decision Letter 1]

22 Dec 2020

PONE-D-20-29297R1

High SARS-CoV-2 load in the nasopharynx of patients with a mild form of COVID-19 is associated with clinical deterioration regardless of the hydroxychloroquine administration

PLOS ONE

Dear Dr.Vasilieva, 

Thank you for submitting your manuscript to PLOS ONE. After careful consideration, we feel that it has merit but does not fully meet PLOS ONE’s publication criteria as it currently stands. Therefore, we invite you to submit a revised version of the manuscript that addresses the points raised during the review process.

We look forward to receiving your revised manuscript.

Kind regards,

Walter R. Taylor

Academic Editor

PLOS ONE

Additional Editor Comments (if provided):

Dear Dr. Vasilieva,

Thank you for submitting your revised paper.

The paper is much improved but there remain a few minor issues I would like you to respond to, as listed below.

Abstract

I suggest you change the text to something like this:

In the current non-randomized, observational study we evaluated the SARS-CoV-2 RNA in nasopharynx swabs from infected patients 7-10 days post symptoms with clinically mild disease and compared the viral RNA load dynamics between patients receiving HCQ (200 mg twice per day according to the Ministry of Health of Russian Federation treatment instructions, n = 33) and a control group without antiviral pharmacological therapy (n = 12).

Introduction

Please change this because ‘highly controversial’ does not apply to all studies:

From:

Furthermore, use of chloroquine and HCQ for SARS-CoV-2 patients with severe disease has been reported, but the results of these studies are highly controversial [4-10]. While there are not enough clinical data to support the use of HCQ, several countries including Russia have included HCQ in treatment protocols for infected patients and for prophylaxis.

To:

Early in the endemic, there were insufficient clinical data to support a particular dose of HCQ so several countries like Russia recommended different HCQ doses as treatment for infected patients and as prophylaxis.

Initial results suggesting that HCQ and CQ had antiviral activity and were beneficial (Lammers, Gao Gautret) were not confirmed in the large Solidarity and Recovery randomized trials of severe COVID-19 and early treatment of mild COVID-19 (Mitja et al, Skipper et al); these trials used higher HCQ doses than the one recommended in Russia.

Please amend your referencing accordingly. The Self reference is less relevant now that we have the Solidarity & Recovery trials so up to you if you wish to cite it.

Current citation for the Solidarity trial.

WHO Solidarity Trial Consortium, Pan H, Peto R, Henao-Restrepo AM, Preziosi MP, Sathiyamoorthy V, Abdool Karim Q, Alejandria MM, Hernández García C, Kieny MP, Malekzadeh R, Murthy S, Reddy KS, Roses Periago M, Abi Hanna P, Ader F, Al-Bader AM, Alhasawi A, Allum E, Alotaibi A, Alvarez-Moreno CA, Appadoo S, Asiri A, Aukrust P, Barratt-Due A, Bellani S, Branca M, Cappel-Porter HBC, Cerrato N, Chow TS, Como N, Eustace J, García PJ, Godbole S, Gotuzzo E, Griskevicius L, Hamra R, Hassan M, Hassany M, Hutton D, Irmansyah I, Jancoriene L, Kirwan J, Kumar S, Lennon P, Lopardo G, Lydon P, Magrini N, Maguire T, Manevska S, Manuel O, McGinty S, Medina MT, Mesa Rubio ML, Miranda-Montoya MC, Nel J, Nunes EP, Perola M, Portolés A, Rasmin MR, Raza A, Rees H, Reges PPS, Rogers CA, Salami K, Salvadori MI, Sinani N, Sterne JAC, Stevanovikj M, Tacconelli E, Tikkinen KAO, Trelle S, Zaid H, Røttingen JA, Swaminathan S. Repurposed Antiviral Drugs for Covid-19 - Interim WHO Solidarity Trial Results. N Engl J Med. 2020 Dec 2: NEJMoa2023184.

Citations for Mitja & Skipper

Mitjà O et al. A Cluster-Randomized Trial of Hydroxychloroquine for Prevention of Covid-19. N Engl J Med. 2020 Nov 24:NEJMoa2021801.

Skipper et al. Hydroxychloroquine in Nonhospitalized Adults With Early COVID-19 : A Randomized Trial. Ann Intern Med.2020 Oct 20;173(8):623-631.

Methods

Line 66: change to: while 12, who declined HCQ, represented a control group.

Conclusion

I suggest changing line 243 to:

At the dose used in this study in mildly ill patients who were symptomatic for at least one week, HCQ did not accelerate viral clearance compared to no HCQ.

Yours sincerely,

Walter Taylor.

---

## [Author Response · Author response to Decision Letter 1]

23 Dec 2020

All authors would like to thank the Academic Editor for the suggested text edits. The amendments introduced are listed below:

Response to the Academic Editor’s comments:

Abstract:

“Abstract

I suggest you change the text to something like this:

In the current non-randomized, observational study we evaluated the SARS-CoV-2 RNA in nasopharynx swabs from infected patients 7-10 days post symptoms with clinically mild disease and compared the viral RNA load dynamics between patients receiving HCQ (200 mg twice per day according to the Ministry of Health of Russian Federation treatment instructions, n = 33) and a control group without antiviral pharmacological therapy (n = 12).”

The text was modified according to the comment (Page 2, Line 28).

“Introduction

Please change this because ‘highly controversial’ does not apply to all studies:

From:

Furthermore, use of chloroquine and HCQ for SARS-CoV-2 patients with severe disease has been reported, but the results of these studies are highly controversial [4-10]. While there are not enough clinical data to support the use of HCQ, several countries including Russia have included HCQ in treatment protocols for infected patients and for prophylaxis.

To:

Early in the endemic, there were insufficient clinical data to support a particular dose of HCQ so several countries like Russia recommended different HCQ doses as treatment for infected patients and as prophylaxis.

Initial results suggesting that HCQ and CQ had antiviral activity and were beneficial (Lammers, Gao Gautret) were not confirmed in the large Solidarity and Recovery randomized trials of severe COVID-19 and early treatment of mild COVID-19 (Mitja et al, Skipper et al); these trials used higher HCQ doses than the one recommended in Russia.

Please amend your referencing accordingly.”

Corresponding edits and referencing were introduced in accordance with suggested text variant (Page 3, Lines 43-49).

“Methods

Line 66: change to: while 12, who declined HCQ, represented a control group.”

The corresponding sentence was modified according to the comment (Page 4, Line 67).

“Conclusion

I suggest changing line 243 to:

At the dose used in this study in mildly ill patients who were symptomatic for at least one week, HCQ did not accelerate viral clearance compared to no HCQ.”

The text was modified according to the comment (Page 12, Lines 238-240).

---

## [Editor Report · Decision Letter 2]

18 Jan 2021

High SARS-CoV-2 load in the nasopharynx of patients with a mild form of COVID-19 is associated with clinical deterioration regardless of the hydroxychloroquine administration

PONE-D-20-29297R2

Dear Dr. Vasilieva,

We’re pleased to inform you that your manuscript has been judged scientifically suitable for publication and will be formally accepted for publication once it meets all outstanding technical requirements.

Kind regards,

Walter R. Taylor

Academic Editor

PLOS ONE
---

## [Editor Report · Acceptance letter]

21 Jan 2021

PONE-D-20-29297R2 

High SARS-CoV-2 load in the nasopharynx of patients with a mild form of COVID-19 is associated with clinical deterioration regardless of the hydroxychloroquine administration 

Dear Dr. Vasilieva:

I'm pleased to inform you that your manuscript has been deemed suitable for publication in PLOS ONE. Congratulations! Your manuscript is now with our production department. 

Kind regards, 

on behalf of

Dr. Walter R. Taylor 

Academic Editor

PLOS ONE